# Rethinking the Number of Shots in Robust Model-Agnostic Meta-Learning

## Abstract

Robust Model-Agnostic Meta-Learning (MAML) is usually adopted to train a meta-model which may fast adapt to novel classes with only a few exemplars and meanwhile remain robust to adversarial attacks. The conventional solution for robust MAML is to introduce robustness-promoting regularization during meta-training stage. However, although the robustness can be largely improved, previous methods sacrifice clean accuracy a lot. In this paper, we observe that introducing robustness-promoting regularization into MAML reduces the intrinsic dimension of clean sample features, which results in a lower capacity of clean representations. This may explain why the clean accuracy of previous robust MAML methods drops severely. Based on this observation, we propose a simple strategy, *i.e.*, setting the number of training shots larger than that of test shots, to mitigate the loss of intrinsic dimension caused by robustness-promoting regularization. Though simple, our method remarkably improves the clean accuracy of MAML without much loss of robustness, producing a robust yet accurate model. Extensive experiments demonstrate that our method outperforms prior arts in achieving a better trade-off between accuracy and robustness. Besides, we observe that our method is less sensitive to the number of fine-tuning steps during training, which allows for a reduced number of fine-tuning steps to improve training efficiency.

## 1 Introduction

Few-shot learning (Finn et al., 2017; Rajasegaran et al., 2020; Li et al., 2021; Dong et al., 2022) aims to train a model which can fast adapt to novel classes with only a few exemplars. Model-agnostic meta-learning (MAML) (Finn et al., 2017) is a typical meta-learning approach to deal with few-shot learning problems. However, the model trained through MAML is not robust to adversarial attacks. The conventional adversarial training can facilitate MAML with adversarial robustness. However, the limited data in the few-shot setting makes it challenging (Goldblum et al., 2020) to keep both clean accuracy and robustness at a high level at the same time.

In recent years, a series of works (Yin et al., 2018; Goldblum et al., 2020; Wang et al., 2021) pay attention to the robust MAML. Most of them attempt to introduce robustness-promoting regularization (*i.e.*, adversarial loss) into the typical MAML bi-level training framework, *e.g.*, AQ (Goldblum et al., 2020), ADML (Yin et al., 2018), and R-MAML (Wang et al., 2021). Although those methods introduce robustness-promoting regularization in different ways, all of them follow the typical MAML training practice that the number of training shots should match with the number of test shots. For plain MAML (*i.e.*, without robustness-promoting regularization), this practice is optimal to achieve the best novel class adaptation performance (Finn et al., 2017; Cao et al., 2019), and setting the number of training shots different from that of test shots will harm the performance of plain MAML. For example, as shown in Fig. 1(b), for 5-way 1-shot testing tasks on miniImageNet (Vinyals et al., 2016), plain MAML achieves the highest clean accuracy when the number of training shots is the same with that of test shots (*i.e.*, 1). As we increase the training shot number, its clean accuracy decreases. However, when we empirically evaluate the performance of the aforementioned robust MAML methods (*i.e.*, AQ, ADML and R-MAML), two interesting phenomenons are observed: **1)** We fairly compare the performance of these robust MAML methods, and find that compared to plain MAML, all of them greatly sacrifice the clean accuracy to improve their robustness (see Fig. 1(a)). **2)** We also observe that when we make the number of training shots larger than that of test shots, the clean accuracy of robust MAML shows a different pattern from plain MAML. Specifically, for the 5-way 1-shot testing task shown in Fig. 1(b), unlike plain MAML, the clean

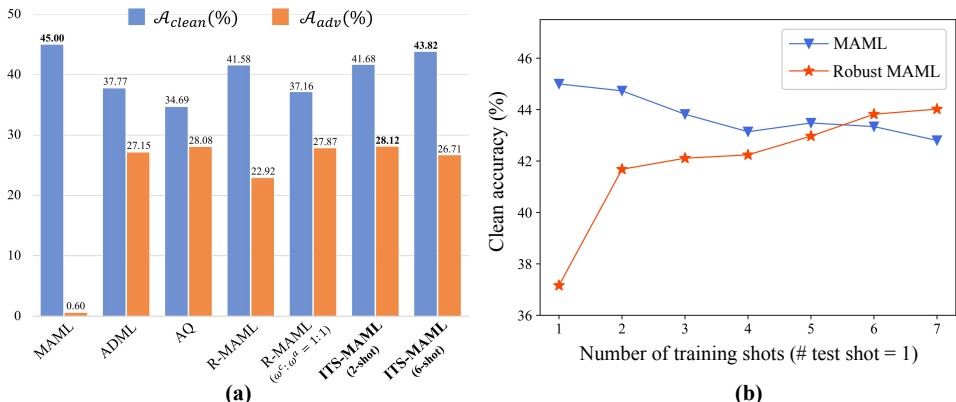

(a)                 (b)

Figure 1: **(a)** Accuracy of models trained by MAML, ADML, AQ, R-MAML and our ITS-MAML on 5-way 1-shot miniImageNet (Vinyals et al., 2016) testing tasks. The $\mathcal{A}_{clean}$ and $\mathcal{A}_{adv}$ denote clean accuracy and robust accuracy, respectively. We adopt a 10-step PGD attack (Madry et al., 2017) with power $\epsilon = 2$. The $w^c$ and $w^a$ are the weights of clean loss and adversarial loss, respectively. The original setting of R-MAML is equivalent to $w^c : w^a = 1 : 0.2$ for higher clean accuracy. We also report the results of R-MAML with $w^c : w^a = 1 : 1$ for a fair comparison with other methods. **(b)** Clean accuracy of models trained by plain MAML and robust MAML (Wang et al., 2021) with different number of training shots on 5-way 1-shot miniImageNet (Vinyals et al., 2016) testing task.

accuracy of robust MAML *increases* instead of *decreasing* as we increase the training shot number. These phenomenons imply that the conventional MAML practice, which sets the training and test shot numbers the same, may not be optimal for robust MAML. Thus, a natural question arises: comparing plain MAML and robust MAML, what is the underlying factor that causes the different patterns of clean accuracy as the number of training shots changes?

In deep CNNs, intrinsic dimension (ID) of feature embedding, which is defined as the minimum variations captured to describe a representation or realize a specific optimization goal, has proved to be an accurate predictor of the network's classification accuracy or generalization ability on the test set (Ansuini et al., 2019; Cao et al., 2019; Aghajanyan et al., 2020). Recall that the optimization objective in plain MAML framework is to maximize the likelihood on clean query images, which can be roughly viewed as maximizing the generalization ability to unseen data in each episode. The ID of feature embedding in plain MAML actually represents the minimum variations needed to guarantee the generalization ability in each few-shot task. Thus, to enable the model to achieve a maximum generalization ability or optimal testing performance, the effective principle for MAML training should be *making the underlying ID of the training stage approximately match with that of the testing stage*. In plain MAML, such a principle can be realized via setting the number of training shots to that of testing shots. However, the practice working for plain MAML may not work for robust MAML. Given the same number of training shots, introducing adversarial loss into MAML framework (*i.e.*, robust MAML) may disturb the normal feature learning and result in a lower ID of clean features than the plain MAML framework. This may explain why introducing adversarial loss into MAML framework may obviously sacrifice the clean accuracy, as introducing adversarial loss reduces the required ID, thus resulting in a lower capacity of clean representations.

In this paper, based on our observations, we propose a simple way, *i.e.*, *setting the number of training shots larger than that of test shots*, to mitigate the loss of ID of clean features caused by robustness-promoting regularization. For example, if we consider the $N$-way 1-shot task, we may use a larger number of training shots in robust MAML, *i.e.* larger than 1. We denote our method as "ITS-MAML" as it requires **I**ncreasing the **T**raining **S**hot number. Such a design is inspired by the observation that increasing the number of training shots may yield the increased ID of clean representations Cao et al. (2019). Through setting the number of training shots larger than that of test shots, we expect the gap of affordable ID between training and testing can be mitigated. With this simple operation, our method improves clean accuracy of robust MAML remarkably without much loss of robustness, as shown by the results of ITS-MAML (with the training shot number set to 2 or 6) in Fig. 1(a). Extensive experiments on miniImageNet (Vinyals et al., 2016), CIFAR-FS (Bertinetto et al., 2018), and Omniglot (Lake et al., 2015) demonstrate that our method performs favourably against previous robust MAML methods considering both clean accuracy and robustness. We also

show that our method achieves a better trade-off between clean accuracy and robustness. Finally, we demonstrate that compared to previous robust MAML methods, our method is less sensitive to the number of fine-tuning (inner-loop) steps in meta-training, and bares almost no drop of accuracy even when the number of fine-tuning steps is greatly reduced, thus improving training efficiency.

In a nutshell, our contributions are summarized as: **1)** We observe that introducing robustness-promoting regularization into MAML reduces the intrinsic dimension of features, which means the capacity of representations may be largely reduced. This implies the conventional MAML practice that the number of training shots should match with the number of test shots is not optimal for robust MAML setting. **2)** Based on our observations, we propose a simple yet effective strategy, *i.e.*, setting the number of training shots larger than that of test shots, to mitigate the loss of intrinsic dimension caused by robustness-promoting regularization. **3)** Extensive experiments on three few-shot learning benchmarks, *i.e.*, miniImageNet (Vinyals et al., 2016), CIFAR-FS (Bertinetto et al., 2018), and Omniglot (Lake et al., 2015), demonstrate that our method performs favourably against previous robust MAML methods considering both clean accuracy and robustness. We also demonstrate that our method can achieve a better trade-off between clean accuracy and robustness and may improve the training efficiency by reducing the fine-tuning steps at the meta-training stage.

## 2 RELATED WORK

**Adversarial robustness.** Adversarial robustness refers to the accuracy of the model on adversarially perturbed samples, which are visually indistinguishable from clean samples but can drastically change the model predictions (Ilyas et al., 2019; Xie & Yuille, 2019). Adversarial training is one of the most effective approaches to improve the model's adversarial robustness (Goodfellow et al., 2014; Madry et al., 2017; Zhang et al., 2019b). For example, Goodfellow et al. (2014) proposed for the first time to adopt single-step-based adversarial samples for adversarial training, and (Madry et al., 2017) further extended it to multi-step-based adversarial examples for better robustness. Others propose methods for fast adversarial training to improve the training efficiency (Shafahi et al., 2019; Zhang et al., 2019a; Wong et al., 2020). Methods have been proposed to improve the transferablity of robustness in few-shot settings (Chen et al., 2020; Chan et al., 2020; Tian et al., 2020; Rizve et al., 2021). However, learning an adversarially robust meta-model is challenging. The improvement of robustness is often accompanied by the decline of accuracy since a fundamental trade-off between the clean and adversarial distributions exists, as is theoretically proved by (Zhang et al., 2019b). The scarce data in few-shot settings makes it harder to learn the robust decision boundary, resulting in even more vulnerability of the model against adversarial attacks (Xu et al., 2021).

**Adversarially robust model-agnostic meta-learning.** Meta-learning has demonstrated promising performance in few-shot learning (Snell et al., 2017; Huang et al., 2018; Maicas et al., 2018; Sung et al., 2018; Wang et al., 2020). MAML (Finn et al., 2017), as the first to propose an effective meta-learning framework, can learn a well-initialized meta-model to quickly adapt to new few-shot classification tasks. Though widely adopted, MAML naturally lacks adversarial robustness. A few work studied the robustness of MAML including ADML (Yin et al., 2018), AQ (Goldblum et al., 2020) and R-MAML (Wang et al., 2021). However, these methods do not achieve a satisfactory trade-off between clean accuracy and adversarial robustness. For example, AQ trades accuracy for robustness compared with ADML, while R-MAML achieves a high clean accuracy on condition that the robustness is greatly reduced compared with AQ (see Fig. 1(a)). In addition, all these methods greatly sacrifice clean accuracy compared with plain MAML.

## 3 METHODOLOGY

### 3.1 PRELIMINARY

**MAML for few-shot learning.** Few-shot learning aims to enable the model to classify data from novel classes with only a few (*e.g.*, 1) examples to train. Few-shot learning is typically formulated as a $N$-way $K$-shot classification problem, where in each task, we aim to classify samples (denoted as "query") into $N$ different classes, with $K$ samples (denoted as "support") in each class for training. Meta-learning is the conventional way to deal with the few-shot learning problem, while model-agnostic meta-learning (MAML) is one of the most popular and effective meta-learning methods.

Generally speaking, MAML attempts to learn a well-initialized model (*i.e.*, a *meta-model*), which can quickly adapt to new few-shot classification tasks. During meta-training, the meta-model is fine-tuned over $N$ classes (with each class containing $K$ samples), and then updated by minimizing

Figure 2: Robust meta-learning frameworks of previous arts and our method. $\theta$ denotes the initial model parameters in each episode, $\theta'$ denotes the model parameters after fine-tuning on the support set (inner-loop), and $\mathcal{L}$ is the loss of the fine-tuned model on the query set, adopted for the meta-update of $\theta$ (outer-loop). $\mathcal{S}$ and $\mathcal{Q}$ are the support data and the query data, with the superscripts $c$ and $a$ denoting clean and adversarial samples respectively. For a $N$-way $K$-shot meta-testing task, our method simply sets the training shot number $\tilde{K}$ larger than $K$. Note that each episode consists of several few-shot classification tasks. We omit the task index in our symbols for brevity.

the validation error of the fine-tuned network over unseen samples from these $N$ classes. The *inner fine-tuning* stage and the *outer meta-update* stage form the *bi-level* learning procedure of MAML. Formally, we consider $T$ few-shot classification tasks, each of which contains a support data set $\mathcal{S} = \{s_1, s_2, \cdots, s_K\}$ ($s_j \in \mathcal{S}$ denotes the $j$-th support sample) for fine-tuning, and a query data set $\mathcal{Q}$ for meta-update. MAML's bi-level optimization problem can be described as:

$$\underset{\theta}{\text{minimize}} \frac{1}{T} \sum_{i=1}^{T} \mathcal{L}_{\theta'_i}(\mathcal{Q}_i),$$

$$\text{subject to } \theta'_i = \arg\min_{\theta} \mathcal{L}_{\theta}(\mathcal{S}_i), \ \forall i \in \{1, 2, \ldots, T\}, \tag{1}$$

where $\theta$ is the meta-model to be learned, $\theta'_i$ is the fine-tuned parameters for the $i$-th task, $\mathcal{L}_{\theta}(\mathcal{S}_i)$ and $\mathcal{L}_{\theta'_i}(\mathcal{Q}_i)$ are the training error on the support set and the validation error on the query set, respectively. Note that the fine-tuning stage (corresponding to the constraint in Eq. 1) usually calls for $M$ steps of gradient update:

$$\theta_i^{(m)} = \theta_i^{(m-1)} - \alpha \nabla_{\theta_i^{(m-1)}} \mathcal{L}_{\theta_i^{(m-1)}}(\mathcal{S}_i), \ m \in \{1, 2, \ldots, M\}, \tag{2}$$

where $\alpha$ is the learning rate of the inner update, $\theta_i^{(0)} = \theta$ and $\theta_i^{(M)} = \theta'_i$.

**Robust MAML.** Adversarial training is one of the most effective defense methods to learn a robust model against adversarial attacks (Madry et al., 2017). Suppose $\mathcal{D} = \{\mathcal{D}^c, \mathcal{D}^a\}$ denotes the set of samples used for training, and the adversarial training can be represented as

$$\underset{\theta}{\text{minimize}} \quad w^c \cdot \mathcal{L}_{\theta}(\mathcal{D}^c) + w^a \cdot \mathcal{G}_{\theta}(\mathcal{D}^a), \tag{3}$$

where $\theta$ is the parameters of robust model to be learned, $\mathcal{L}_{\theta}(\mathcal{D}^c)$ is the prediction loss (*e.g.*, cross-entropy loss) on clean sample set $\mathcal{D}^c$, $\mathcal{G}_{\theta}(\mathcal{D}^a)$ is the adversarial loss (*e.g.*, cross entropy loss (Goodfellow et al., 2014; Madry et al., 2017) or KL divergence (Zhang et al., 2019b)) on adversarial sample set $\mathcal{D}^a$, and $w^c$ and $w^a$ are the weights of clean loss and adversarial loss respectively. Each adversarial sample $x^a \in \mathcal{D}^a$ is obtained by adding perturbations to the clean sample $x^c \in \mathcal{D}^c$ to maximize its classification loss:

$$x^a = x^c + \arg\max_{\|\delta\|_p \leq \epsilon} \mathcal{G}_{\theta}(x^c + \delta), \tag{4}$$

where the $p$-norm of the perturbation $\delta$ is limited within the $\epsilon$ bound so that the adversarial samples are visually indistinguishable from the clean samples.

Existing robust MAML methods introduced adversarial loss to MAML's bi-level learning procedure. As illustrated in Fig. 2, AQ (Goldblum et al., 2020) directly replaced the loss on clean query images of plain MAML with adversarial loss. Compared to AQ, ADML (Yin et al., 2018) additionally added another optimization pathway, *i.e.*, fine-tuning with the adversarial loss on support data and evaluating the accuracy on clean query data. Further, R-MAML (Wang et al., 2021) showed that there is no need to impose adversarial loss during the fine-tuning stage and imposed both the clean prediction loss and the adversarial loss. Although R-MAML demonstrated superior clean accuracy

Table 1: Intrinsic dimensions of features trained by plain MAML and the robust MAML (Wang et al., 2021) with different numbers of training shots. The "C" and "N" denote the intrinsic dimension of clean samples and the adversarial noise (added to clean samples), respectively.

| Method | | Number of training shots | | | | | | |
|---|---|---|---|---|---|---|---|---|
| | | 1 | 2 | 3 | 4 | 5 | 6 | 7 |
| MAML | C | 80 | 103 | 139 | 151 | 157 | 160 | 179 |
| Robust MAML | C | 22 | 41 | 59 | 60 | 71 | 86 | 89 |
| | N | 4 | 7 | 7 | 8 | 8 | 9 | 10 |

compared to ADML, we find that if we treat the clean loss and adversarial loss on query data equally, *i.e.*, setting the learning rate of clean loss to that of adversarial loss, the clean accuracy of R-MAML actually performs comparably to ADML (as shown in Fig. 1(a)). Thus, R-MAML actually improves clean accuracy by sacrificing robustness (*i.e.*, increasing the learning rate of clean loss). There remains a question about how to improve clean accuracy while maintaining good robustness.

## 3.2 Rethinking the number of shots

**Robustness regularization reduces intrinsic dimension of features.** Intrinsic dimension (ID) of features denotes the minimum variations captured to describe the representations. From the optimization objective, it can be viewed as the minimum free factors or parameters needed to realize the optimization objective. Previous works (Ansuini et al., 2019; Aghajanyan et al., 2020) showed that the ID of features is usually smaller than the length of feature vectors (*e.g.*, the number of units in a specific layer of deep neural network). For a $\mathbb{R}^D$ feature space, the ID (denoted as $\hat{D}$) usually satisfies $\hat{D} < D$. In deep CNN, previous work (Ansuini et al., 2019) demonstrated that the ID of last hidden layer performs as an accurate predictor of network's generalization ability. Thus, in this paper, we attempt to analyze the drop of clean accuracy in robust MAML through the lens of ID.

There are different ways to estimate $\hat{D}$. One of the simplest ways is to adopt Principal Component Analysis (PCA) to discover the number of principal variations (Huang, 2018; Cao et al., 2019). In our paper, the ID is set to the number of principal components which retain at least $90\%$ of the variance of the features trained on miniImageNet. Other estimation methods can also be employed (Huang, 2018; Facco et al., 2017; Amsaleg et al., 2015) (see Appendix for more details).

As shown in Table 1, we observe that: **1)** Given the same number of training shots, introducing adversarial loss into MAML framework results in a lower ID of clean samples than the plain MAML framework which doesn't adopt any adversarial loss during training. It may be because unexpected adversarial noise disturbs the normal representation learning on clean data. Recall that the optimization objective in plain MAML is to maximize the likelihood on clean query images, which can be roughly viewed as maximizing the generalization ability to unseen data in each episode. Thus, from the optimization perspective, the ID in plain MAML actually represents the minimum variations needed to guarantee the generalization ability on clean data in each few-shot episode. This may explain why introducing adversarial loss into MAML framework may obviously sacrifice the clean accuracy, as introducing adversarial loss reduces the required ID and lowers the capacity of clean representations. **2)** We also investigate how the ID of adversarial noise changes as the training shot number increases for robust MAML in Table 1. We observe that the ID of adversarial noise for robust MAML is insensitive to the number of training shots, indicating that the adversarial robustness may not be affected by different number of training shots that much.

All these observations demonstrate that in the context of robust MAML, the conventional MAML practice, where the training and test shot numbers should be set the same, may not be optimal.

**Increasing the number of training shots.** From Table 1, we also observe that as the training shot number increases, the ID of clean representations also increases for both plain MAML and robust MAML. Based on these observations, we propose a simple yet effective way, *i.e.*, setting the number of training shots larger than that of test shots, to mitigate the loss of intrinsic dimension caused by

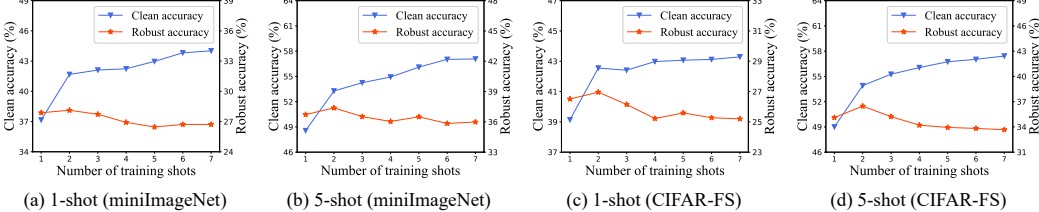

(a) 1-shot (miniImageNet)  (b) 5-shot (miniImageNet)  (c) 1-shot (CIFAR-FS)  (d) 5-shot (CIFAR-FS)

Figure 3: Clean accuracy and robust accuracy of models on 5-way 1-shot and 5-way 5-shot meta-testing tasks. The experiments are conducted on miniImageNet and CIFAR-FS. The models are trained with different numbers of training shots. The robust accuracy is computed with a 10-step PGD attack with $\epsilon = 2$ for miniImageNet and $\epsilon = 8$ for CIFAR-FS.

adding adversarial loss. Formally, for a $N$-way $K$-shot testing task, our method is expressed as

$$\operatorname*{minimize}_{\theta} \frac{1}{T} \sum_{i=1}^{T} w^c \cdot \mathcal{L}_{\theta_i'}(\mathcal{Q}^c) + w^a \cdot \mathcal{L}_{\theta_i'}(\mathcal{Q}^a),$$

$$\text{subject to } \theta_i' = \arg\min_{\theta} \mathcal{L}_\theta(\tilde{\mathcal{S}}_i^c), \tag{5}$$

where $\tilde{\mathcal{S}}_i^c = \{s_1, s_2, \cdots, s_{\tilde{K}}\}$ denotes the support set for the $i$-th task during meta-training. In our method, the number of support images (*i.e.*, training shot number) $\tilde{K}$ for each task at meta-training stage is set to be larger than the number of support images $K$ used in meta-testing. For example, if we deal with the 5-way 1-shot testing, $\tilde{K}$ is set to be larger than 1. The $w^c$ and $w^a$ are the weights of clean loss $\mathcal{L}_{\theta_i'}(\mathcal{Q}^c)$ and adversarial loss $\mathcal{L}_{\theta_i'}(\mathcal{Q}^a)$, respectively. Empirically, we find that $w^c = w^a = 1$ is sufficient to achieve a good trade-off between clean accuracy and robustness.

Though simple, we empirically find that our method can remarkably improve the clean accuracy of the model without much loss of robustness, thus yielding an accurate yet robust model. As shown in Fig. 3, with increasing the number of training shots, the clean accuracy steadily increases before reaching a bound. For the robustness, it only slightly decreases but still remains high, which matches with the phenomenon we observed in Table 1 (*i.e.*, the intrinsic dimension of adversarial noise is quite low and insensitive to the number of training shots).

**Discussion.** A natural question may arise: why is the optimal way of setting the number of training shots different between plain MAML and robust MAML (see Fig. 1(b))? As the ID can be interpreted as representing the minimum variations to guarantee the generalization ability in each few-shot task, we may also explain this phenomenon through the lens of ID: **1)** For plain MAML, considering a $K$-shot testing, using a larger training shot number (*i.e.*, $\tilde{K} > K$) yields larger ID during training than the affordable ID during testing ($K$-shot). This means if we set $\tilde{K} > K$, *the resulting representation capacity of training is larger than what can be afforded by testing data, causing overfitting*. Thus, it is better to keep $\tilde{K} = K$ for plain MAML. **2)** For robust MAML, it is different. Introducing adversarial loss reduces ID compared to plain MAML. Under this circumstance, if we still keep the training shot number $\tilde{K} = K$, the resulting representation capacity of training is lower than what can be afforded by testing data, leading to sub-optimal results. To sum up, the seemingly conflict practice between plain MAML and robust MAML actually follows the same principle, *i.e.*, we should approximately *match* the underlying ID of training with that of testing to maximize the generalization ability.

# 4 EXPERIMENTS

## 4.1 SETUP

**Dataset.** We conduct experiments on three widely-used few-shot learning benchmarks, *i.e.*, miniImageNet (Vinyals et al., 2016), CIFAR-FS (Bertinetto et al., 2018), and Omniglot (Lake et al., 2015). The *miniImageNet* contains 100 classes with 600 samples in each class. The whole dataset is split into 64, 16 and 20 classes for training, validation and testing respectively. We adopt the training set for meta-training, and randomly select 2000 unseen tasks from the testing set for meta-testing. Each image is downsized to $84 \times 84 \times 3$ in our experiments. *CIFAR-FS* has the same dataset splitting as miniImageNet, *i.e.*, 64, 16 and 20 classes for training, validation and testing respectively, with each class containing 600 images. We also adopt the training set for meta-training, and randomly

Table 2: Accuracy of meta-models trained by different methods under different types of attacks on miniImageNet. The best results are marked in bold.

| Method | Model | Training shot number ($\tilde{K}$) | 5-way 1-shot testing acc. (%) ($K=1$) | | | | 5-way 5-shot testing acc. (%) ($K=5$) | | | |
|---|---|---|---|---|---|---|---|---|---|---|
| | | | Clean | FGSM | PGD | CW | Clean | FGSM | PGD | CW |
| MAML | 4-layer CNN | 1 | **45.00** | 3.71 | 0.60 | 0.24 | **58.64** | 7.36 | 1.72 | 2.14 |
| ADML | 4-layer CNN | 1 | 37.77 | 29.96 | 27.15 | 26.79 | 56.02 | 41.96 | 35.90 | 35.66 |
| AQ | 4-layer CNN | 1 | 34.69 | 30.53 | 28.08 | 26.20 | 52.66 | 42.82 | 37.21 | 37.84 |
| R-MAML | 4-layer CNN | 1 | 37.16 | 29.95 | 27.87 | 26.13 | 55.85 | 42.63 | 36.30 | 34.93 |
| **ITS-MAML** | 4-layer CNN | 2 | 41.68 | **31.74** | **28.12** | **27.84** | 53.37 | **43.68** | **37.38** | **38.22** |
| **ITS-MAML** | 4-layer CNN | 6 | **43.82** | 30.60 | 26.71 | 26.34 | **57.03** | 42.81 | 35.84 | 35.40 |
| MAML | ResNet-12 | 1 | **53.27** | 8.00 | 3.00 | 2.46 | **70.18** | 19.22 | 7.18 | 10.28 |
| ADML | ResNet-12 | 1 | 51.32 | 33.53 | 31.30 | 29.96 | 66.08 | 45.30 | 42.10 | 44.73 |
| AQ | ResNet-12 | 1 | 49.36 | 33.91 | 32.40 | 33.06 | 64.85 | 47.82 | **45.46** | 45.98 |
| R-MAML | ResNet-12 | 1 | 50.71 | 33.08 | 31.14 | 30.28 | 67.84 | 47.11 | 44.79 | 45.77 |
| **ITS-MAML** | ResNet-12 | 2 | 52.78 | **34.26** | **32.57** | **33.18** | 67.07 | **47.84** | 45.12 | **46.20** |
| **ITS-MAML** | ResNet-12 | 6 | **53.00** | 33.14 | 30.88 | 30.50 | **68.72** | 46.96 | 43.73 | 45.08 |

select 4000 unseen tasks from the testing set for meta-testing. Each image is resized to $32 \times 32 \times 3$. *Omniglot* includes handwritten characters from 50 different alphabets, with a total of 1028 classes of training data and 423 classes of testing data. We randomly select 2000 unseen tasks from the testing data for meta-testing. Each image has a size of $28 \times 28 \times 1$.

**Training.** We verify our method based on two kinds of architectures, *i.e.*, a four-layer convolutional neural network as in (Wang et al., 2021) and a ResNet-12 (He et al., 2016). All the models in our experiments are trained for 12 epochs unless specified. During meta-training, each episode consists of 4 randomly selected tasks. For 5-way 1-shot meta-testing tasks, the number of support images per class in each task of meta-training is 1 for previous methods, and 2 for ITS-MAML. For 5-way 5-shot meta-testing tasks, the number of support images per class in each task of meta-training is 5 for previous methods, and 6 for ITS-MAML. The number of query images per class is 15 for all methods on miniImageNet and CIFAR-FS. On Omniglot, since each class only contains 20 samples, the number of query images per class is thus set to 9 for 5-way 1-shot meta-testing tasks and 5 for 5-way 5-shot meta-testing tasks to avoid data repetition in a task. The learning rate is set to 0.01 for fine-tuning, and 0.001 for the meta-update. Following (Wang et al., 2021), the number of fine-tuning steps is set to 5 for meta-training, and 10 for meta-testing for all methods unless specified. As (Wang et al., 2021) showed that adopting the FGSM attack (Goodfellow et al., 2014) instead of the PGD attack (Madry et al., 2017) during meta-training can improve the training efficiency without significantly affecting the model performance, we also adopt the FGSM attack (Goodfellow et al., 2014) as the training attack. The training attack power $\epsilon$ is set to 2 for miniImageNet, and 10 for CIFAR-FS and Omniglot. We evaluate our method under different kinds of attacks for testing. The testing attack power is 2 for miniImageNet, 8 for CIFAR-FS and 10 for Omniglot unless specified. For a fair comparison, we set $w^c : w^a = 1 : 1$ in Eq. 5 for all methods unless specified.

**Fair comparison.** For all our experiments, we consider the typical $N$-way $K$-shot setting. In practice, a $N$-way $K$-shot task means at the **test** stage, we need to classify unseen samples into $N$ classes based on $K$ annotated samples per category to fine-tune. In nature, a $N$-way $K$-shot setting doesn't restrict the way how we train our model, which means during training, we don't need to formulate the training task same as the test task. Whether the number of training shots $\tilde{K}$ and that of test shots $K$ are the same just reflects the different designs of different methods, but doesn't mean different settings are considered. Thus, our method doesn't violate the requirements of typical $N$-way $K$-shot tasks.

## 4.2 COMPARISONS WITH PREVIOUS METHODS

We evaluate the performance of our proposed method (denoted as "ITS-MAML") under 5-way 1-shot and 5-way 5-shot settings, and compare our method with plain MAML and previous typical robust MAML methods, *i.e.*, AQ, ADML and R-MAML. We demonstrate the experimental results on three benchmarks in Tables 2, 3 and 4, respectively. In those tables, we show the accuracy on

Table 3: Accuracy of meta-models trained by different methods under different types of attacks on CIFAR-FS. A 4-layer CNN is adopted for all methods.

| Method | Training shot number ($\tilde{K}$) | 5-way 1-shot testing acc. (%) ($K=1$) | | | | 5-way 5-shot testing acc. (%) ($K=5$) | | | |
|---|---|---|---|---|---|---|---|---|---|
| | | Clean | FGSM | PGD | CW | Clean | FGSM | PGD | CW |
| MAML | 1 | **49.93** | 9.57 | 0.15 | 0.08 | **65.63** | 18.18 | 0.70 | 1.22 |
| ADML | 1 | 40.41 | 36.67 | 26.05 | 25.79 | 56.52 | 49.63 | 32.08 | 32.86 |
| AQ | 1 | 32.08 | 30.69 | 26.49 | 23.08 | 51.40 | 48.19 | 35.08 | 33.90 |
| R-MAML | 1 | 39.14 | 35.79 | 26.51 | 25.77 | 56.09 | 50.78 | 32.67 | 34.38 |
| **ITS-MAML** | 2 | 42.55 | **38.52** | **26.96** | **27.60** | 53.91 | **51.46** | **36.47** | **37.21** |
| **ITS-MAML** | 6 | **43.12** | 38.38 | 25.37 | 26.24 | **57.12** | 51.07 | 33.62 | 34.05 |

Table 4: Accuracy of meta-models trained by different methods under different types of attacks on Omniglot. A 4-layer CNN is adopted for all methods.

| Method | Training shot number ($\tilde{K}$) | 5-way 1-shot testing acc. (%) ($K=1$) | | | | 5-way 5-shot testing acc. (%) ($K=5$) | | | |
|---|---|---|---|---|---|---|---|---|---|
| | | Clean | FGSM | PGD | CW | Clean | FGSM | PGD | CW |
| MAML | 1 | 93.02 | 65.20 | 22.72 | 26.64 | 97.78 | 91.75 | 60.25 | 54.39 |
| ADML | 1 | 90.58 | 89.84 | 78.27 | 77.19 | 97.46 | 96.53 | 91.06 | **91.50** |
| AQ | 1 | 90.09 | 88.72 | 81.69 | 80.92 | 97.02 | 96.44 | 91.25 | 90.11 |
| R-MAML | 1 | 89.60 | 87.65 | 77.44 | 77.57 | 97.12 | 96.09 | 90.28 | 89.72 |
| **ITS-MAML** | 2 | 93.56 | **91.43** | **85.45** | **83.68** | 97.02 | **96.80** | **91.87** | 91.46 |
| **ITS-MAML** | 6 | **94.23** | 86.89 | 79.90 | 77.34 | **98.19** | 96.19 | 90.77 | 88.52 |

clean images ("Clean") and robust accuracy under different types of attacks, *i.e.*, FGSM (Goodfellow et al., 2014), PGD (Madry et al., 2017), and CW (Carlini & Wagner, 2017). The accuracy under different types of attacks reflects the model's robustness level. Based on the experiment results, we have three observations: **1)** Compared to MAML, all robust MAML methods (including ours) significantly improve the model's robustness. For example, for 4-layer CNN on 5-way 1-shot task of miniImageNet, compared to MAML baseline, AQ, ADML, R-MAML and ITS-MAML ($\tilde{K} = 2$) improve the accuracy under PGD attack by about 27%, 27%, 27% and 28%, respectively. These results verify that introducing adversarial loss into MAML contributes to the model's robustness. **2)** By comparing previous robust MAML methods (*e.g.*, ADML and R-MAML), we find that generally they achieve comparable results for both clean accuracy and robustness, indicating whether applying the adversarial loss to the fine-tuning procedure doesn't bring a huge difference. For example, for 4-layer CNN on 5-way 1-shot testing of miniImageNet, the clean accuracy for ADML and R-MAML is 37.77% and 37.16% respectively, while the robust accuracy is 27.15% and 27.87% (under PGD attack). Generally, AQ performs worse than ADML and R-MAML on clean accuracy, which indicates imposing clean loss for meta-update may be necessary for a high clean accuracy. **3)** Compared to previous robust MAML methods, our strategy of increasing training shot number (ITS-MAML) remarkably improves the clean accuracy while maintaining good robustness (sometimes the robustness is even better than previous methods). For example, on 5-way 1-shot task on CIFAR-FS, in terms of clean accuracy, ITS-MAML ($\tilde{K} = 2$) outperforms AQ and R-MAML by more than 10% and 3%. If we increase $\tilde{K}$ from 2 to 6, we observe that ITS-MAML ($\tilde{K} = 6$) further improves the clean accuracy by around 1%. The robust accuracy decreases slightly, but still remains at a high level, which is competitive among previous robust MAML methods. For 5-way 5-shot testing, we observe the similar trend. Thus, if we care more about the model's robustness, we may lower $\tilde{K}$. However, if we want to achieve a robust yet accurate model, increasing $\tilde{K}$ is a good choice.

### 4.3 Ablation Study and Analysis

**Better trade-off between clean accuracy and robustness.** R-MAML (Wang et al., 2021) demonstrated superior clean accuracy compared to AQ and ADML, when the learning rate of clean loss is set to 5 times the learning rate of adversarial loss, *i.e.*, the ratio $w^c : w^a$ is set to $1 : 0.2$ (see Eq. 5). We find that when this ratio is set to be consistent with ADML and ITS-MAML, *i.e.*, $1 : 1$, the high clean accuracy of R-MAML no longer exists (see Fig. 1(a)). It implies that there exists a trade-off between clean accuracy and robustness with the change of this ratio. We compare our ITS-MAML

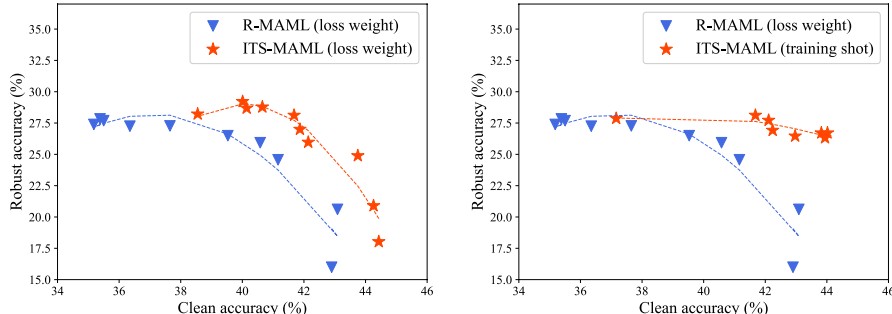

Figure 4: Clean *vs.* robust accuracy. **Left**: Varying $w^c : w^a$ from $1 : 0.1$ to $1 : 5$ for R-MAML and ITS-MAML ($\tilde{K} = 2$). **Right**: Varying $w^c : w^a$ for R-MAML and $\tilde{K}$ for ITS-MAML ($w^c : w^a = 1 : 1$).

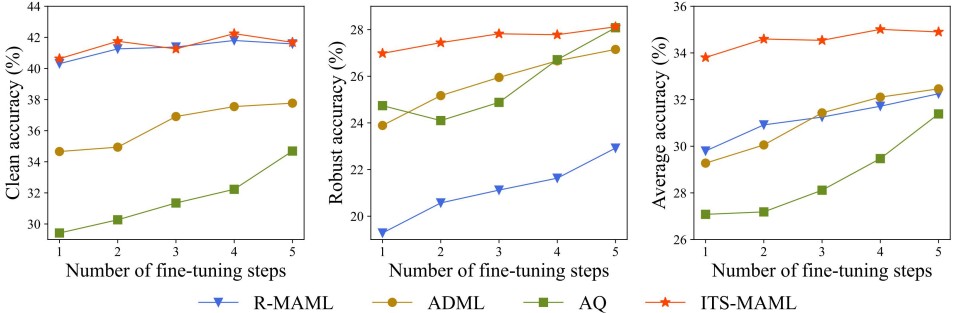

Figure 5: Accuracy of models trained by ADML, AQ, R-MAML ($w^c : w^a = 1 : 0.2$ as in (Wang et al., 2021)) and ITS-MAML with different number of fine-tuning steps during meta-training. **Left**: clean accuracy. **Middle**: Robust accuracy, which is computed with a 10-step PGD attack with $\epsilon = 2$. **Right**: Average accuracy, which denotes the average value of clean accuracy and robust accuracy.

with R-MAML under different ratios of $w^c : w^a$ and show corresponding clean accuracy and robustness in Fig. 4. To demonstrate the effectiveness of setting $\tilde{K}$ larger than $K$, we also show the results of ITS-MAML under different $\tilde{K}$ (keeping $w^c : w^a = 1 : 1$) in Fig. 4. From Fig. 4 (left), we observe that the larger the ratio of $w^c : w^a$, the higher the clean accuracy and the lower the robust accuracy of the model. Compared with R-MAML, ITS-MAML achieves obviously better trade-off (the curve of ITS-MAML is above that of R-MAML). In addition, Fig. 4 (right) demonstrates that as $\tilde{K}$ increases, the clean accuracy of ITS-MAML steadily increases before reaching a bound, while the robustness remains relatively stable at a high level, which further verifies the better trade-off by our method.

**More efficient training.** The number of fine-tuning steps (*i.e.*, $M$ in Eq. 2) during meta-training affects the training efficiency of the model. Larger $M$ leads to higher computation costs and a lower training efficiency. We expect that ITS-MAML can achieve good performance even if the number of fine-tuning steps $M$ is reduced during training, thus improving the training efficiency. To this end, we train ADML, AQ, R-MAML ($w^c : w^a = 1 : 0.2$ as in (Wang et al., 2021)) and ITS-MAML models with $M$ ranging from 1 to 5. We then evaluate the 5-way 1-shot accuracy of the models on miniImageNet. The results are shown in Fig. 5. We observe that previous methods may suffer from an obvious performance degradation (either clean accuracy or robustness) with reducing $M$. Compared to previous methods, ITS-MAML is less sensitive to $M$ during meta-training. This allows for a reduced number of fine-tuning steps to improve the training efficiency.

## 5 CONCLUSION

In this paper, we observe that introducing adversarial loss into MAML framework reduces the intrinsic dimension of features, which results in a lower capacity of representations of clean samples. Based on this observation, we propose a simple yet effective strategy, *i.e.*, setting the number of training shots larger than that of test shots, to mitigate the loss of intrinsic dimension caused by introducing adversarial loss. Extensive experiments on few-shot learning benchmarks demonstrate that compared to previous robust MAML methods, our method can achieve superior clean accuracy while maintaining high-level robustness. Further, empirical studies show that our method achieves a better trade-off between clean accuracy and robustness, and has better training efficiency. We hope our new perspective may inspire future research in the field of robust meta-learning.

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

# A APPENDIX

## A.1 DETAILS ABOUT HOW TO ESTIMATE THE INTRINSIC DIMENSION

Intrinsic dimension of features denotes the minimum variations captured to describe the representations. For a $\mathbb{R}^D$ feature space, the intrinsic dimension $\hat{D}$ usually satisfies $\hat{D} < D$. In this section, we provide details about how we estimate the intrinsic dimension of features trained by plain MAML (Finn et al., 2017) and robust MAML. Specifically, we trained 4-layer CNNs with either plain MAML or robust MAML on miniImageNet (Vinyals et al., 2016), with the training shot number varying from 1 to 7. Then, for each trained meta-model, we obtain the feature embeddings of all clean samples in the meta-training set (denoted as $\mathbf{z}^c \in \mathbb{R}^{N \times D}$) and the feature embeddings of the corresponding adversarial noise (denoted as $\mathbf{z}^a \in \mathbb{R}^{N \times D}$), where $N$ is the total number of meta-training samples, and $D$ is the feature dimension. For $\mathbf{z}^c$ and $\mathbf{z}^a$, we take the output of the feature extractor (*i.e.*, the embedding before the classifier). The adversarial noise is generated by FGSM (Goodfellow et al., 2014) with the attack power $\epsilon = 2$.

We adopt two algorithms to estimate the intrinsic dimension of the clean features $\mathbf{z}^c$ and the noise features $\mathbf{z}^a$, *i.e.*, PCA (Cao et al., 2019) and TwoNN (Facco et al., 2017). For PCA (Cao et al., 2019), by eigendecomposing the covariance matrix of embeddings, we obtain the principal components expressed as the significant eigenvalues, and the principal directions expressed as the eigenvectors corresponding to those eigenvalues. We approximate the intrinsic dimension of the embedding space by the number of significant eigenvalues, which is determined by an explained-variance over total-variance criterion, *i.e.*, $r_{\hat{D}} = \frac{\sum_{i \in [1,\hat{D}]} \lambda_i}{\sum_{i \in [1,D]} \lambda_i}$, where the threshold $r_{\hat{D}}$ is set to 0.9 in our experiments. The result for PCA estimation is provided in the paper.

We also adopt TwoNN (Facco et al., 2017) algorithm to estimate the intrinsic dimension. TwoNN estimates the intrinsic dimension by a minimal neighborhood information. Specifically, TwoNN estimator employs only the distances to the first and the second nearest neighbor of each point, which is designed to mitigate the influence of feature inhomogeneities on the estimation process. In our experiments, we discard 10% of the points characterized by highest values of $l_2/l_1$, where $l_1$ and $l_2$ are the distances to the first and the second nearest neighbor of a point, respectively. The result for TwoNN estimation is provided in Table 5. From the result, we observe a similar trend to the result of PCA estimation: 1) Given the same number of training shots, introducing adversarial loss into MAML framework results in a lower intrinsic dimension of clean samples than the plain MAML, leading to a lower capacity of clean representations. 2) The intrinsic dimension of adversarial noise is insensitive to the training shot number, indicating that the adversarial robustness may not be affected by the variation of training shot number that much. 3) With increasing the number of training shots, the intrinsic dimension of clean representations also increases for both MAML and robust MAML.

Table 5: Intrinsic dimensions of features trained by plain MAML and the robust MAML (Wang et al., 2021) with different numbers of training shots. The intrinsic dimension is estimated through TwoNN (Facco et al., 2017) of features trained on miniImageNet. The "C" and "N" denote the intrinsic dimension of clean samples and the adversarial noise (added to clean samples), respectively.

| Method | | Number of training shots | | | | | | |
| --- | --- | --- | --- | --- | --- | --- | --- | --- |
| | | 1 | 2 | 3 | 4 | 5 | 6 | 7 |
| MAML | C | 37.6 | 40.1 | 42.9 | 43.7 | 45.3 | 45.7 | 46.4 |
| Robust MAML | C | 21.6 | 24.1 | 26.1 | 26.3 | 27.4 | 27.5 | 28.7 |
| | N | 4.4 | 6.1 | 5.4 | 5.6 | 6.2 | 6.4 | 5.8 |

## A.2 VERIFICATION ON OTHER META-LEARNING ALGORITHMS

Following previous works (Goldblum et al., 2020; Yin et al., 2018; Wang et al., 2021), we initially focus on the MAML algorithm. By examining our motivation and findings, we believe that our method also works with other meta-learning algorithms. Therefore, we conduct experiments with another meta-learning algorithm, *i.e.*, ProtoNet (Snell et al., 2017). Specifically, following the same

settings of plain MAML, R-MAML and ITS-MAML respectively, we train plain ProtoNet (without adversarial loss), R-ProtoNet ($\tilde{K} = 1$ with adversarial loss), and ITS-ProtoNet ($\tilde{K} = 2$ with adversarial loss), except that the meta-learning algorithm is changed from MAML to ProtoNet. A ResNet-18 (He et al., 2016) is adopted for all methods. The results provided in Table 6 below show similar trend to those of MAML. The "ProtoNet" which is trained without adversarial loss achieves the highest clean accuracy (45.94%), but lowest robustness. The "R-ProtoNet" which is trained with adversarial loss achieves good robustness, but sacrifices clean accuracy a lot. By increasing $\tilde{K}$ from 1 to 2, our method greatly increases the clean accuracy (41.44% vs. 44%) and slightly increases the robustness across different attacks. This demonstrates the effectiveness and generalization of our method.

Table 6: Accuracy of meta-models trained by different methods under different types of attacks on miniImageNet. ProtoNet (Snell et al., 2017) is adopted as the meta-learning algorithm. The best results are marked in bold.

| Method | Training shot number ($\tilde{K}$) | 5-way 1-shot testing acc. (%) ($K=1$) | | | |
|---|---|---|---|---|---|
| | | Clean | FGSM | PGD | CW |
| ProtoNet | 1 | **45.94** | 4.67 | 3.28 | 3.65 |
| R-ProtoNet | 1 | 41.44 | 25.77 | 24.11 | 23.92 |
| **ITS-ProtoNet** | **2** | **44.00** | **26.41** | **24.60** | **25.07** |

### A.3 ROBUSTNESS TO OTHER TYPES OF DISTRIBUTION SHIFT

Same with previous robust MAML methods (Goldblum et al., 2020; Yin et al., 2018; Wang et al., 2021), we initially focus on studying the robustness under adversarial attacks. To demonstrate that the proposed method can also provide robustness to other types of distribution shift, we also study the natural noise case following the same training and testing setting as that in the experiment part. We choose three types of typical noises (*i.e.*, Impulse Noise, Motion Blur and Snow) from the benchmark ImageNet-C (Hendrycks & Dietterich, 2019). A ResNet-18 (He et al., 2016) is adopted for all methods. The results in Table 7 show a similar trend to the adversarial noise case, *i.e.*, introducing robustness-promoting regularization into the training framework improves the model's robustness to the noises, but greatly sacrifices its accuracy on clean samples. Setting the number of training shots larger than that of test shots mitigates the gap in clean accuracy while maintaining high robustness to the natural noises.

Table 7: Accuracy of meta-models trained by different methods under different types of natural noises from ImageNet-C. The best results are marked in bold.

| Method | Training shot number ($\tilde{K}$) | 5-way 1-shot testing acc. (%) ($K=1$) | | | |
|---|---|---|---|---|---|
| | | Clean | Impulse | Motion | Snow |
| MAML | 1 | **52.25** | 25.40 | 29.91 | 39.76 |
| R-MAML | 1 | 49.58 | 37.91 | 39.00 | 45.89 |
| **ITS-MAML** | **2** | **51.36** | **41.64** | **39.76** | **47.76** |

### A.4 COMPLEXITY ANALYSIS

We also analyze the time and memory complexity of the proposed method to demonstrate that the extra computational burden introduced by increasing the training shot number is affordable.

Compared to the complexity increase caused by introducing adversarial loss into the MAML framework, the complexity increase caused by *increasing $\tilde{K}$ by 1* is smaller and acceptable. For example, for a 4-layer CNN (Wang et al., 2021), with the training shot number $\tilde{K}=1$, adding adversarial loss

into MAML increases the training time by 387s per epoch (*i.e.*, 801s vs. 414s), while increasing the training shot number $\tilde{K}$ from 1 to 2 only increases the training time by 13s (*i.e.*, 427s vs. 414s). The memory burden shows a similar trend. For example, with $\tilde{K} = 1$, introducing adversarial loss into MAML increases the memory burden by 1108MiB (*i.e.*, 4906MiB vs. 3798MiB), while increasing $\tilde{K}$ from 1 to 2 only increases the memory burden by 568MiB (*i.e.*, 4366MiB vs. 3798MiB). Therefore, we may increase $\tilde{K}$ NOT too high if we care more about complexity. For example, setting $\tilde{K} = 2$ for 1-shot testing (or setting $\tilde{K} = 6$ for 5-shot testing) yields considerable performance gains with an affordable complexity increase. Further optimization on complexity remains future work.

