# OpenReview forum: "Rethinking the Number of Shots in Robust Model-Agnostic Meta-Learning"
_ICLR.cc/2024/Conference — ICLR 2024 Conference Withdrawn Submission_

### Official Review · Reviewer_u51s · 2023-10-31

**Soundness:** 3 good
**Presentation:** 3 good
**Contribution:** 3 good
**Rating:** 5
**Confidence:** 3

**Summary:**

This paper investigates the effect of the number of shots during meta-training for robust meta-learning based on the model R-MAML. It proposes that increasing the number of shots during meta-training to be larger than that of meta-testing could improve the clean accuracy and without much sacrifice of robust accuracy. In addition, this paper explains this observation using the intrinsic dimension of features. Through extensive experiments, the paper validates their claims.

**Strengths:**

1. The motivation is clear and strong.
2. The idea is simple and effective.
3. The proposed approach is explained appropriately using the concept of the intrinsic dimension of features.

**Weaknesses:**

1. The concept of intrisic dimension (ID) of features could be used to explain the reason why increasing the number of shots could improve the clean accuracy during meta-testing. However, it might be counterintuitive for the corresponding decreasing robust accuracy. For example, in Table 1, we find that the ID also increase for adversarial noise when increasing the number of training shots. According to the logic for clean accuracy, the robust accuracy should also increase?
2. Even if the ID could explain the effect of enlarging training shots, the interpretation is still from a high-level perspective (although I am not very familiar with ID). That would be great if the interpretation could be from quantitative analysis in detail.

**Questions:**

As we know, transfer-learning based or model pre-trained based approaches outperform meta-learning based approaches for few-shot learning. Also, enlarging the number of training shots for MAML could decrease MAML's performance. The author also explained this through ID. I am curious that whether ID could be used to explain why meta-learning approaches underperform transfer-learning based approach. Is that possible to show the ID analysis for a plain transfer-learning based approach, such as [1] or other related papers.

[1] A Closer Look at Few-shot Classification, ICLR 2019

From an extreme perspective, enlarging the number of training shots to a big number, we can view this is close to a pretrained model based on all meta-training dataset? If so, the meta-test performance of transfer-learning based approach should underperform meta-learning based approach.

I am open to improve my rating based on the response.

---

### Official Review · Reviewer_fxbE · 2023-11-01

**Soundness:** 2 fair
**Presentation:** 2 fair
**Contribution:** 1 poor
**Rating:** 3
**Confidence:** 4

**Summary:**

The author notes a reduction in the intrinsic dimension of features from MAML (Model-agnostic meta-learning) when integrated with adversarial training, attributing this to a discrepancy between meta-training and meta-testing, resulting in diminished clean accuracy in prior methodologies. The document puts forth a straightforward remedy: augmenting the number of shots during meta-training. Through empirical evidence, the author demonstrates that this approach effectively elevates the intrinsic dimension of the features, enhancing the clean accuracy of the few-shot learning model, all the while preserving robust accuracy.

**Strengths:**

1. The paper presents a straightforward method to enhance compromised clean accuracy.

2. The paper is well-organized and written in an easily understandable manner.

**Weaknesses:**

1. The explanation for why the adversarial loss reduces the Intrinsic Dimensionality (ID) is lacking, and it remains unconvinced that increasing the number of training shots effectively enhances generalization or not. It is imperative to confirm that the increase in variance is not solely a result of having more training shots.

2. It has not been explicitly stated whether Cross-Entropy (CE) or Kullback-Leibler (KL) divergence was employed as the adversarial loss. If CE was utilized, this represents a deviation from the previous approach's loss setting, and if KL was used, the 1:1 weight ratio differs from the previous configuration. Irrespective of the chosen loss function, it is essential to provide an explanation and rationale for the divergence in optimization methodology employed in the prior approaches.

3. The evaluation setting appears to deviate from the norm. While a training epsilon of 10 and a testing epsilon of 8 were utilized, the existing baseline models (AQ and RMAML) seem to employ an epsilon value of 8 for both training and testing. It is necessary to elucidate this disparity and also disclose the alpha value that was set.

4. Given that the proposed method of increasing the training shots is orthogonal, it is imperative to validate whether this approach yields similar effects across ADML, AQ, and RMAML. A comparative analysis to assess the impact of increased training shots on these different methods is warranted.

**Questions:**

NA

---

### Official Review · Reviewer_biDU · 2023-11-02

**Soundness:** 3 good
**Presentation:** 3 good
**Contribution:** 3 good
**Rating:** 5
**Confidence:** 4

**Summary:**

- Introduction of the "ITS-MAML" method that improves clean accuracy without sacrificing robustness.

- Extensive empirical validation on benchmarks like miniImageNet, CIFAR-FS, and Omniglot.

- Demonstration that the method is less sensitive to the number of fine-tuning steps, improving training efficiency.

The core concept revolves around the "intrinsic dimension" of feature embeddings. A higher intrinsic dimension generally means that the model has a greater capacity to represent complex data structures. The authors argue that robustness-promoting regularization in MAML reduces this intrinsic dimension, leading to a decrease in clean accuracy. The authors propose a simple yet effective strategy to tackle this issue: Increase the number of "training shots" (samples) used during the meta-training phase compared to the number of "test shots" used during the meta-testing phase. The idea is that by using more training shots, the model can potentially increase the intrinsic dimension of the feature embeddings for clean samples. This, in turn, would allow the model to maintain high clean accuracy without sacrificing robustness a lot.

**Strengths:**

Originality

The paper introduces a novel approach, "ITS-MAML," to address the trade-off between robustness and clean accuracy in Model-Agnostic Meta-Learning (MAML). The idea of manipulating the number of training shots to affect the intrinsic dimension of feature embeddings is relatively unique. This approach diverges from the more common techniques of introducing new architectures or loss functions to achieve the same goal. Therefore, in terms of originality, the paper appears to be quite strong.

Quality

The quality of the paper is decent. The authors back their claims with extensive experiments on well-known benchmarks like miniImageNet, CIFAR-FS, and Omniglot. They also compare their method with existing robust MAML methods, demonstrating its efficacy.

Clarity

The paper is well-structured and clearly written. It starts by identifying the problem, provides a theoretical background, and then moves on to describe the proposed method and experimental setup. The use of mathematical formulations and graphs also aids in understanding the concepts better.

**Weaknesses:**

- the paper could be strengthened by including more diverse datasets or real-world applications to validate the generalizability of the approach.

- the paper could benefit from a more detailed explanation or visualization of how the intrinsic dimension is affected by the number of shots.

- The paper could strengthen its claims by testing the method on more diverse datasets or in practical scenarios.

- how does ITS-MAML fare against non-MAML methods that also aim for a balance between robustness and accuracy

- A more rigorous theoretical analysis or proof could strengthen the paper's claims.

- ITS-MAML is less sensitive to the number of fine-tuning steps, but it doesn't provide an in-depth sensitivity analysis for other hyperparameters like learning rates, regularization terms, etc. Understanding how sensitive the method is to various hyperparameters could provide more insights into its robustness and generalizability.

**Questions:**

see weaknesses

---

### Official Review · Reviewer_bLFF · 2023-11-10

**Soundness:** 3 good
**Presentation:** 2 fair
**Contribution:** 2 fair
**Rating:** 3
**Confidence:** 4

**Summary:**

In this paper, the authors have tried to rethink the discrepancy in the number of shots between training and test episodes when aiming to train robust meta-learning. Based on the theoretical understanding of the number of shots in meta-learning (Cao et al., 2020), the authors intentionally impose the increased shots in training to match the intrinsic dimension (ID), which is found to be suppressed when training with a robustness regularization term. The authors have confirmed that the intentional discrepancy of shots keeps the clean accuracy while acquiring robustness capability for meta-learning.

**Strengths:**

**Strength 1:** The proposed strategy is very simple to apply, so I feel that readers can easily reconstruct the results and anticipate the applications to other meta-learning methods. I think there is no non-trivial hardship to applying the method to any episode-based few-shot learners.

**Strength 2:** It is widely known that achieving both robustness and clean performance is non-trivial. The work focuses on this issue, especially for the meta-learning topic, via employing a simple yet effective approach that can be easily plugged in with the existing methods. When seeing the experimental results, the authors seem to achieve both goals with the well-kept clean accuracies and the largely-improved robustness performance.

**Strength 3:** This paper is quite well-written, with a clear motivation and reasoning for their proposed method.

**Weaknesses:**

**Weakness 1:** The main concern is that the main contribution remains technical advancement rather than providing the theoretical relationships between the number of shots and the robustness objectives. For the most relevant prior work by (Cao et al., 2019), the theoretical analysis of the number of shots with a lens of VC dimensions is provided. Specifically, Cao et al., provide the lower bound of risk with the terms of shots and statistics of the class prototypes. When stepping forward beyond the prior work, a thorough mathematical investigation should be done to unveil the clear relationship between shots and robustness. To the best of my understanding, I feel that the understanding of shots and robustness remains at the same level of intuitions of the prior work and relies on the empirical, not theoretical, evidence that the robustness regularization term suppresses the models' intrinsic dimension.

**Weakness 2:** This work is based on the optimization-based meta-learning framework, i.e., MAML. However, what happens for the distance metric-based approaches such as Prototypical Nets? In fact, the most related work by (Cao et al., 2019) considers metric-based approaches when examining the effect of shots on the bound of risk. However, this work focuses on optimization-based approaches, which are fairly different from the metric-based methods. When seeing the prior work by (Goldblum et al., 2020), the feature representation of these two branches of methods is fairly different from each other. Therefore, I am concerned about how the message from (Cao et al., 2019) is consistent with the analysis of this work, even with the difference between the two branches of meta-learning.

There are some minor corrections, including typos:
- On page 5: "doesn't" -> "does not"
- On page 7: "doesn't" -> "does not", "don't" -> "do not"
- On page 8: "doesn't" -> "does not"

**Questions:**

**Q1:** For 'Weakness 1', would you provide a further theoretical analysis of the relationships between shots and the robustness? Also, what is the main novelty beyond the prior work by (Cao et al., 2019)?

**Q2:** When experimenting with Prototypical Nets, is the message consistent with the reported results in the submitted version of this paper? Is there any discrepancy between the MAML branch and the Prototypical Nets branch in perspective of the effects of shots on the intrinsic dimension?